# Antimicrobial Coatings: Reviewing Options for Healthcare Applications

Ajay Jose , Marija Gizdavic-Nikolaidis and Simon Swift *

Department of Molecular Medicine and Pathology, Faculty of Medical and Health Sciences, Waipapa Taumata Rau University of Auckland, Auckland 1023, New Zealand
* Correspondence: s.swift@auckland.ac.nz

**Abstract:** Many publications report coatings that exhibit antimicrobial potency applicable to high-touch surfaces and desirable for healthcare settings to contribute to reductions in the occurrence of Hospital Acquired Infections (HAI). In this review, the importance of surface contamination and the transmission of microbes is addressed. The standard strategy to tackle fomites is to implement proper disinfection and cleaning practices and periodically monitor the environment's cleanliness. However, the probability of recontamination of cleaned surfaces is high. Therefore, an additional first line of defense against pathogen transmission and subsequent infection is the antimicrobial surface that can eliminate or at least repel pathogens, introducing a barrier to the spread of infection. It is a simple concept, but formulating a durable, antimicrobial coating with broad-spectrum antimicrobial and antifouling activities has proven challenging. The challenges and progress made in developing such material are reviewed.

**Keywords:** hospital-acquired infections (HAIs); fomites; transmission; antimicrobial coatings; antifouling coatings; durable coatings

## 1. The Problem: HAIs and the Transmission of Pathogens by Fomites

Hospital surfaces, especially high-touch surfaces, are constantly exposed to potentially infectious pathogens. Surfaces or inanimate objects that can carry and potentially transmit pathogens to humans are termed fomites [1,2]. Numerous epidemiological studies exhibit the role of fomites as an essential reservoir of pathogens in transmitting microbes that cause infections in a hospital environment [3]. Viral, bacterial, and fungal pathogen transmission is often more pronounced in hospitals and occurs through two main routes. The primary mode of transmission is direct, human-to-human transmission, where the disease-causing organism is transmitted from an infected patient or carrier to a healthy or susceptible individual. An alternate indirect route of transmission involves fomites that become contaminated with the pathogen and transmit it to others via the contact [4]. Unfortunately, the role of surface-mediated transmission has not received much attention since being first discounted [3–7]. At the beginning of the COVID-19 outbreak, surface transmission was considered the primary route [8–13], and there was substantial further interest in the application of antiviral surfaces that could interrupt transmission [14–16].

The problem of HAIs has been present for centuries (Figure 1) and remains a persistent challenge for healthcare facilities worldwide. HAIs are conditions that patients acquire in a healthcare facility while receiving treatment. HAIs have become a common complication of hospitalization and pose a particular threat to higher-risk groups, including immunocompromised patients, children, HIV patients, patients undergoing surgery, and patients coming to the hospital with open wounds or burns [17]. HAIs can be responsible for extended hospital stays, long-term disability, and preventable deaths, and substantially increase the expenses related to healthcare [17,18], challenging patient safety and the quality of care in all healthcare settings [17]. The rise in the number of immunocompromised patients and our aging population makes HAIs an unavoidable public health challenge [19].

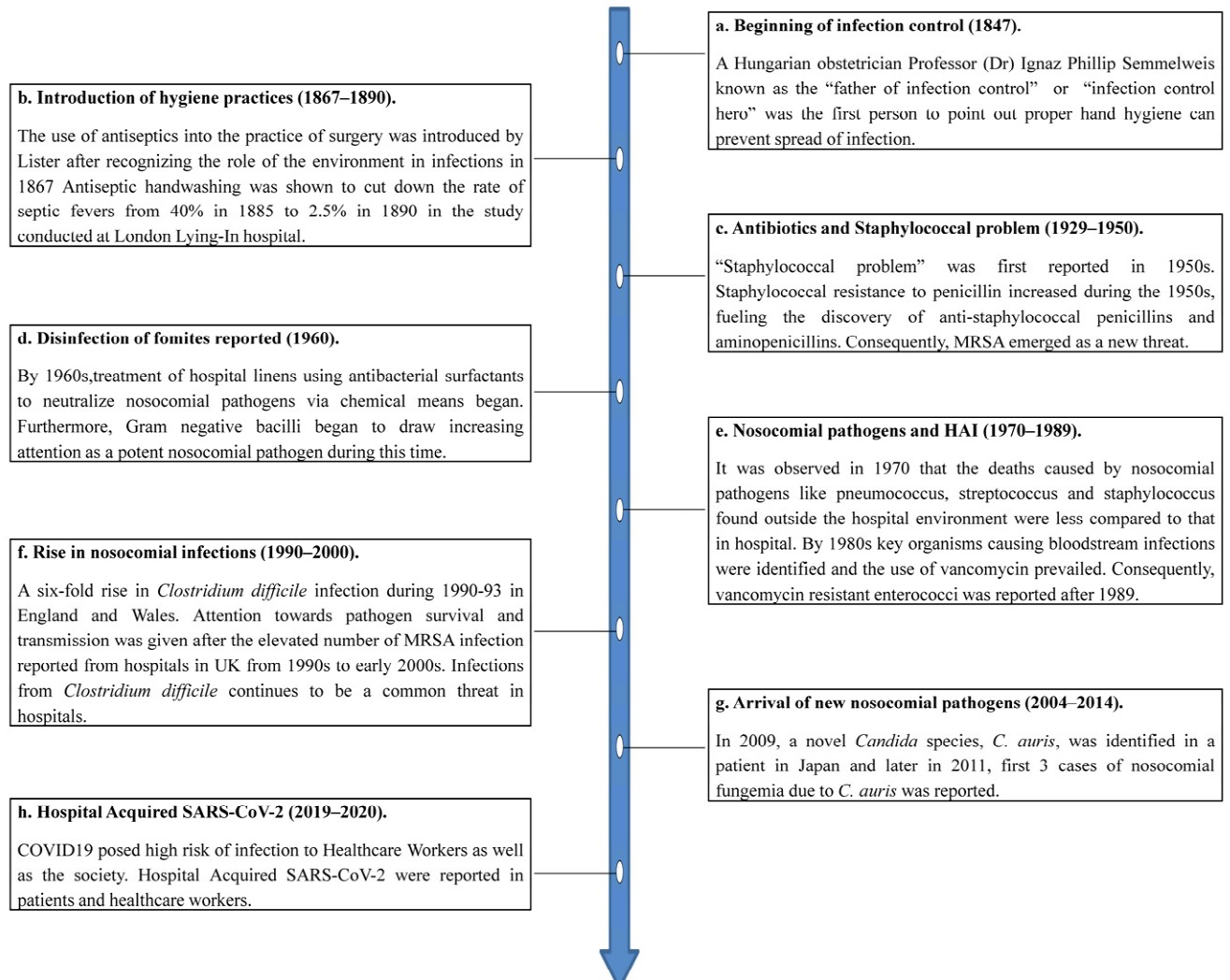

**Figure 1.** Historical overview of HAIs. References: a. [20–22], b. [22,23], c. [22,24,25], d. [26,27], e. [26,28], f. [3,29,30], g. [31,32], and h. [33–35]. It can be observed that the threat is still present, with a clear potential for morbidity and mortality.

Nosocomial pathogens are the microbes that cause HAIs. The two important characteristics of nosocomial pathogens are (i) the capability to cause death if not treated and (ii) to survive on hospital surfaces long enough for transmission to occur [7]. The potential for becoming a nosocomial pathogen is evaluated by the virulence exhibited by the microorganism [36,37]. Unfortunately, the extensive use of antibiotics in healthcare and farming has driven the evolution of multidrug-resistant pathogens that present more troublesome HAIs to confront [37,38]. So, drug-resistant pathogens, such as methicillin-resistant *Staphylococcus aureus* (MRSA), further enhance the risk of HAIs, and new approaches to tackle this growing health burden and its consequences are demanded [19,37,39]. One approach is a durable and reliable solution to interrupt fomite-mediated transmission.

## 2. The Evidence for Surface and Fomite-Mediated Transmission in Healthcare Facilities

Transmission of pathogens is central to the HAI chain [4,17], but understanding the mechanism and contribution of fomite-mediated transmission of nosocomial pathogens to the risk of developing HAIs requires evaluation of complex interactions and interventions within the healthcare settings [4]. Numerous publications have highlighted the role of contaminated environments in the transmission of nosocomial pathogens, including multidrug-resistant organisms [40–42], HAI outbreaks associated with MRSA [43], vancomycin-resistant *Enterococcus* (VRE) [44,45], and *Clostridioides difficile* [46]. If not controlled, the HAI cycle continues until the carriers infect the healthy individuals inside the healthcare premises, as illustrated in Figure 2.

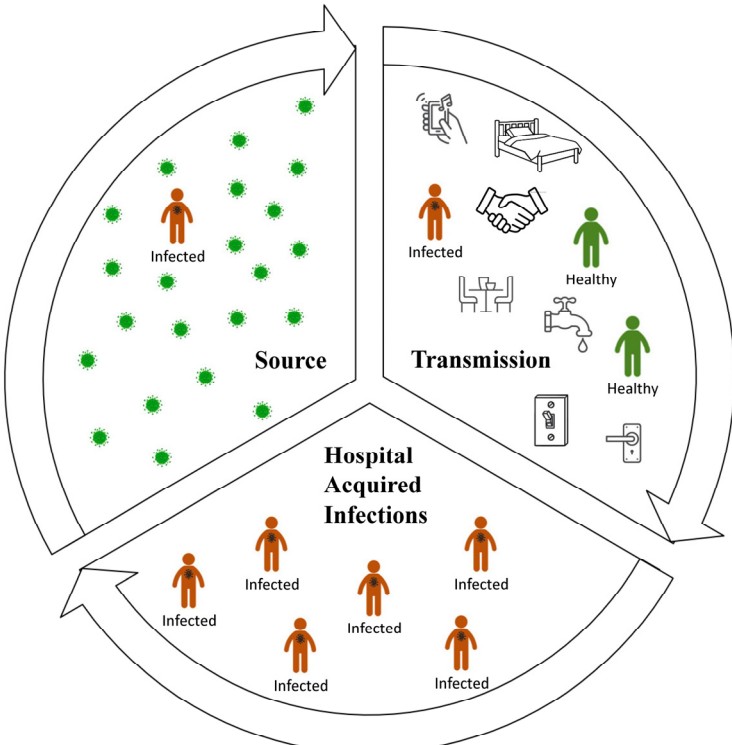

**Figure 2.** Transmission of HAIs in a healthcare facility. Source: Contaminated surfaces, foodstuff, contaminated hands, saliva, respiratory droplets, faeces and other body fluids, contaminated dust or aerosols, and dispersed skin scales [7,36,47–51]. Transmission: Directly by patient-to-patient contact or contact with carrier healthcare workers; indirectly by fomites including surfaces via contact with patients or via healthcare workers and visitors [3,7,52–56]. HAIs: People inside healthcare facilities are the main reservoir, source of pathogens, main transmitter, and simultaneously, the receptor of pathogens and daily activities inside hospitals enables this cycle to continue [7].

Epidemiological studies of nosocomial pathogens show that contaminated surfaces serve as a breeding hub, ultimately resulting in an outbreak [4]. If not cleaned according to the WHO standards, fomites can give rise to biofilms, and it has been established recently that dry surface biofilms (DSB), formed on dry fomites found in a hospital environment [57], are more resistant to cleaning and heat [58–60]. A laboratory study of DSB associated with fomites supported the proposed contribution of DSB in the nosocomial pathogen transmission [61]. *Staphylococcus aureus* DSB grown on polycarbonate and glass surfaces ($10^5$–$10^6$ cfu/coupon) was effectively transferred by the ungloved forefinger, with a single touch transferring around 5% of cells from the fomite to the finger, and 20% of the cells passed on from the finger to an agar plate [61]. In the real world, a study carried out with 61 previously clean inanimate objects from three different United Kingdom hospitals, using scanning electron microscopy (SEM), ribosomal RNA intergenic spacer analysis

(RISA) polymerase chain reaction, and next-generation sequencing identified the presence of diverse dry biofilms on 95% of the samples [62]. Therefore, the chances of nosocomial pathogens recontaminating surfaces even after cleaning and disinfection or the persistence of pathogens toward cleaning are substantial [63]. Despite the findings of these studies, regular analysis for pathogens on surfaces is often viewed as an unnecessary burden and not usually funded [19].

### 3. How Can Fomite-Mediated Transmission Be Managed?

Inside a hospital environment, the fomites containing highly persistent and high-risk pathogens cannot be easily differentiated from clean surfaces. Hence, conventional cleaning and routine disinfection is the primary strategy adopted to defend against environmental contamination and HAIs. A suitable standard operating procedure for hospital cleaning with respect to different wards and surfaces is published by the Centre for Disease Control and Prevention (CDC) and the Infection Control Africa Network (ICAN) [64]. A national standard for cleaning hospitals has been proposed by National Health Service as a re-vised healthcare cleaning manual, and this is strictly implemented in UK hospitals [65]. However, these practices do not effectively eradicate pathogens without properly targeted maintenance [66–68].

It is important to select the suitable cleaning agent for hospitals, a process that is best guided by standards, such as ASTM E2871–21, which determine the efficacy of liquid disinfectants against bacterial biofilms [69]. Using this approach, the common disinfectants, quaternary ammonium compounds (QACs or quats) and sodium hypochlorite (bleach), and other antimicrobials, including phenolics, peracetic acid, and accelerated liquid hydrogen peroxide ($H_2O_2$), have been proven to clean hospital surfaces efficiently [70].

However, disinfectants may fail to deliver the expected performance if not used at the correct concentration for an adequate contact time, leaving viable pathogens on a visibly clean surface [70]. Furthermore, reports showed that conditions exist where bleach may not be effective against *C. difficile* spores [71–73], MRSA [66], or norovirus [74]. Additionally, present-day use of disinfectants (at high concentrations) is limited by their potential for toxicity to staff, damage to materials and equipment, and inactivation by organic matter. Prolonged survival on surfaces and the resistance of potential pathogens to cleaning and liquid disinfectants gradually increases the probability of infection and the risks of an outbreak [75].

Other disinfectants, including ozone [3], vaporized $H_2O_2$ [71], and steam [3], are effective against MRSA, *Escherichia coli*, *C. difficile*, *Enterococcus faecalis*, VRE, *Streptococcus pyogenes*, *Acinetobacter baumannii*, and norovirus after 1–2 h. However, the use of these disinfectants requires vacant rooms, is difficult to handle, and may cause damage to surfaces and devices. Disinfection using germicidal light may offer options for the future. Ultraviolet C (UV-C light, 280 nm) [76–78], pulsed xenon UV light (PX-UV; 200–280 nm) [76,79,80], and high-intensity narrow-spectrum light (HINS; 405 nm) [3,81], all offer efficient killing of *S. aureus*, including MRSA, *Staphylococcus epidermidis*, *Pseudomonas aeruginosa*, *Acinetobacter*, *S. pyogenes*, *Clostridium perfringens*, *E. faecalis*, and VRE within ~10–30 min for germicidal UV wavelengths and 2–5 h for HINS. However, line-of-sight issues, concerns over safety, and the time taken for visible light inactivation limit the use of these technologies.

The limitations of conventional approaches to the cleaning and disinfection of surfaces have led to the investigation of self-disinfecting materials for surface coatings in healthcare facilities in addition to traditional cleaning routines.

### 4. Can Special Surfaces/Coatings Help to Solve the Fomite Problem?

The concept of surfaces that kill or repel pathogens has been around for many years. Foul-resistant coatings developed for marine ships with copper oxide, arsenic, mercury oxide, and organotin derivatives to prevent biofouling have pioneered research in this field [82]. In the mid-1960s, coatings with biocidal tributyl tin (TBT) compounds proved highly effective but were removed from use as their high toxicity to marine life emerged [82].

In general, surfaces that repel pathogens and prevent attachment are termed antifouling, while surfaces that kill microorganisms that attach or come too close are termed antimicrobial [3,83]. In many cases, the antifouling or antimicrobial properties of the surface are created by the application of a coating. The antimicrobial coating (AMC) is biocidal, whereas the antifouling coating (AFC) prevents the attachment of microorganisms to the surface (see Figure 3). Coatings suitable for hospitable surfaces and fomites should have the following properties: non-toxic to the environment, non-toxic to humans, cost-effective, commercially available, lethal to potential pathogens, antifouling, stable and durable. A self-cleaning feature may be desirable to allow the removal of material from dead cells that can interfere with the microbicidal mechanism. The demand for AMC in the healthcare setting began to gain attention as a control measure for HAIs and especially antimicrobial-resistant HAIs, and most recently as a strategy to prevent the fomite-mediated spread of COVID-19 [4]. The topic of antimicrobial surfaces and coatings for the purpose of reducing HAIs has been reviewed by others, covering antifouling, biocide presentation, and release options for self-disinfection [3,84–87] and focusing on the potential of new polymers and rechargeable antimicrobial chemistries [88,89]. In this review, we provide updated classifications, with examples, of the variety of approaches to produce antimicrobial and antifouling surfaces. We highlight the limitations of some approaches and, finally, focus on the potential for surfaces that can provide sustainable and rechargeable activity for long-lasting protection.

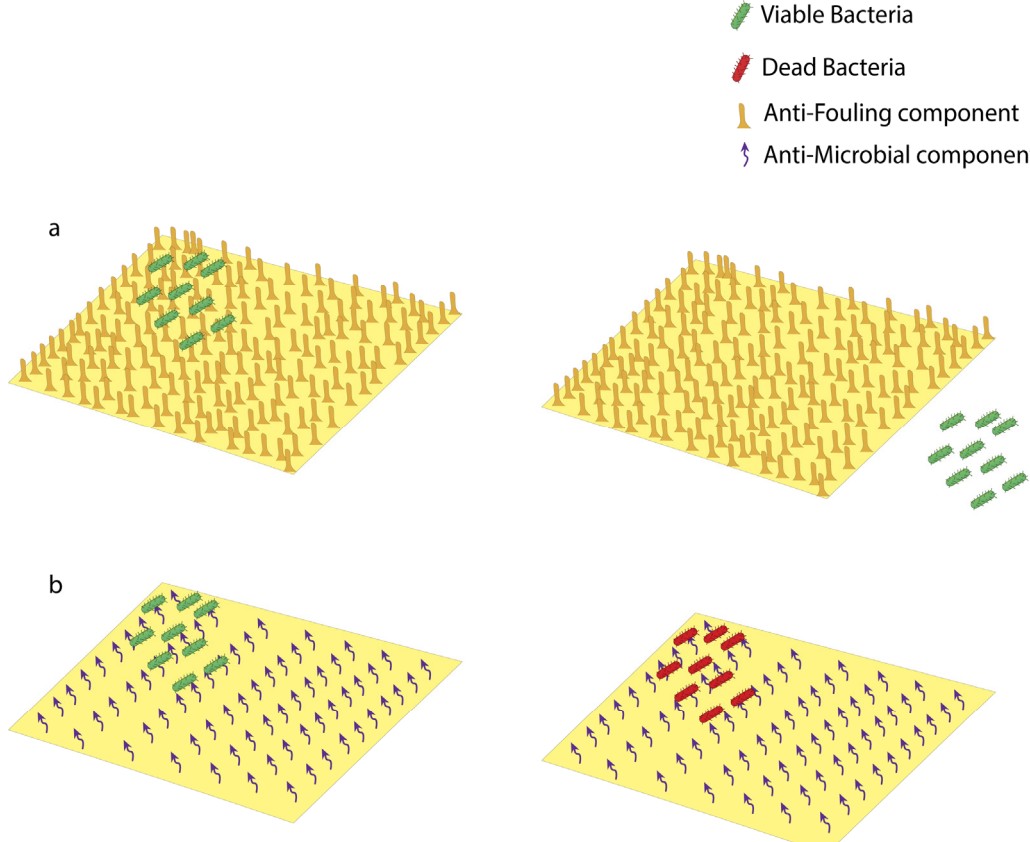

**Figure 3.** Antifouling and antimicrobial surfaces. (**a**) Antifouling surface to block the attachment of pathogens; on the left bacteria approach the surface, on the right the anti-fouling surface blocks attachment. (**b**) Anti-microbial surface to kill pathogens upon contact; on the left bacteria approach the surface, on the right interaction with the anti-microbial surface is lethal.

Before discussing coatings for hospitals further, we need to classify high-touch surfaces in a hospital as large and small objects or areas. This is to identify the suitable coating for each object based on availability, manufacture cost, and required/expected antimicrobial efficiency. In this review, we try to classify them as coatings for large-scale applications (all high-touch surfaces) or coatings suitable for small surfaces. Large-scale surfaces include hospital walls, floors, windows, doors, and all textiles (e.g., curtains, bed sheets, etc.). Small-scale surfaces consist of medical implants, such as catheters, stents, pacemakers, dental implants, and other objects, such as patient care items, clipboards, paper charts, etc. [90–92]. Some coatings can be used for both types of surfaces, while others are specific for objects. Hence, it is important to know the application range for any coating reported. However, most papers fail to report its accurate application, i.e., whether it can be applied on a large object/area or its application is limited to small specific surfaces, such as medical implants, devices, and equipment.

Based on its application, we need to address whether the material chosen for coating repels microbes, kills microbes, or perhaps does both. In addition, it is important to determine whether the coatings are porous or non-porous, rough or smooth, easily applied, and inexpensive. The selection of a method to assess the antimicrobial activity of AFC and AMC based on their application is crucial and should mimic their expected real-world working conditions. For example, it needs to be assessed if they work in wet and/or dry conditions, in the presence of proteinaceous soil, and with a reasonable level of liquid disinfection. Therefore, we consider some key standards and acceptable methods from international journals to study the activity of coatings (both AFC and AMC) for use in hospitals (Table 1).

**Table 1.** Methods for studying the activity of coatings.

| Type of Coating/Surface | Method | Summary | Advantages/Disadvantages |
|---|---|---|---|
| Plastics and other non-porous surfaces | JIS Z 2801 or ISO 22196 [93,94] | Inoculate a known amount of bacteria on a surface. The inoculum is covered using a sterile plastic square piece to ensure uniform spreading and to avoid evaporation of the inoculum. After incubation, the system is transferred to a known volume of selected wash solution, and the surviving bacteria are enumerated by colony counts. The result is interpreted as a log reduction in colony count with respect to control obtained the next day after incubating the plates at 37 °C. | Widely accepted standard to test the antimicrobial potency of coatings. During the washing step, the active biocides may leach out and kill the microbes causing errors in the result. This can be prevented by using neutralizers for the biocide; however, it is dependent upon the concentration of neutralizer, concentration of leachates and specificity of the neutralizer. Some studies report that this method does not reflect real-world conditions of temperature and humidity [95]. It is limited to hard non-porous surfaces only. |
| Copper alloys and silver containing surfaces | Dry fomite assay [96] | A known amount of pathogen is inoculated on the sample surface, and this is incubated under 22 °C and 50% RH for different time intervals, with the surface drying with time. Viable cells removed from the surface by application of a wash solution allow effectiveness to be quantified in comparison to a control surface. | In this method, we test the antimicrobial efficacy under low levels of temperature and humidity to replicate indoor conditions. The RH value is >90% for JIS Z 2801 assay, which may be called as a wet fomite test, while this test ensures a more real-world humidity. |
| Hard, non-porous copper containing surfaces | EPA assay [97] | The Environmental Protection Agency (EPA), US proposed this interim protocol to study the continuous antimicrobial efficacy of a copper containing surface. The protocol is similar to JIS Z 2801; however, it specifies the periodic chemical exposure and mechanical abrasion of the surface. After six weeks of applied wear and tear, the antimicrobial activity of the Cu coating is tested for a period of 2 h. This protocol may be adopted to study the durability of coatings for indoor purposes. | Antibacterial activity monitored with regular cycles of physical abrasion and chemical treatment. Potentially applicable to other non-porous surfaces. |
| Photocatalytic coatings | ISO 27447 [98], ISO 18071 [99], ISO 18061 [100], ISO 13125 [101] | This standard applies to all photocatalytic coatings. The assay is similar to JIS Z 2801 but conducted under two illumination conditions. The sample is treated with the pathogen and allowed to incubate under dark and light conditions (ultraviolet or visible light of known intensity and wavelength) simultaneously. After the illumination period, viable cells are recovered from the surface and enumerated as described in JIS Z 2801 assay. | The standards provide a protocol for testing the efficacy of photocatalytic coatings against bacteria, viruses and fungi. A protocol to test activity under dry conditions is not given. |

**Table 1.** *Cont.*

| Type of Coating/Surface | Method | Summary | Advantages/Disadvantages |
|---|---|---|---|
| Textile products/surfaces | ISO 20743 [102] | This standard specifies three inoculation methods: absorption, transfer, and printing method. In the absorption method, the sterile sample is inoculated with a known amount of bacteria, and the system is treated immediately with 20 mL of wash solution, with shaking for 18–24 h at 37 °C. The final concentration of bacteria is determined using the colony count method. Activity compares cells recovered at time zero and after incubation. The transfer method inoculates the sample by touch transfer from an agar plate with recovery of viable cells at time zero and after incubation for 18–24 h at 37 °C. Activity compares cells recovered at time zero and after incubation. In the printing method, a known amount of bacteria is filtered onto a membrane. Bacteria are transferred to the sample by pressing the test piece on the filter using a weight and rotating 180°. The rest of the procedure follows as described in transfer method. | In some cases, this standard should be combined with other protocols depending on the material used. For example, a textile impregnated with photocatalytic material may need a modified protocol combining both ISO 20743 and ISO 27477 assay. |
| Surfaces, where substrates (fiber, fabric or other substrate) bonded with antimicrobial agents. | ASTM E2149 [103] | This method is used to determine the activity of a sample immersed and shaken in a concentrated bacterial suspension for 1 h. Surviving bacteria are enumerated by colony counting, with activity measured after comparison of viable colonies recovered at time zero and after 1 h. The experiment is simultaneously performed using an appropriate control. | This assay can be modified to assess the antibacterial activity of coatings and thin films [104,105]. |
| Antifouling surfaces | Immersion inoculation assay [95,104,106,107] | The sample is suspended or immersed in a known amount of bacteria. After the required incubation period, the growth media is carefully removed using a sterile pipette and the samples are rinsed to remove residual broth with phosphate buffered saline. The bacteria remaining on the surface are determined using a colony count method or microscopy. | This test can be used to identify the repelling or antifouling activity of a coating under wet conditions [95]. |

**Table 1.** *Cont.*

| Type of Coating/Surface | Method | Summary | Advantages/Disadvantages |
|---|---|---|---|
| Hydrophobic micropattern surfaces | Touch transfer and swab inoculation assay [95,107] | This is reported as the best method for determining anti-attachment and antibacterial activity of nano or micro patterned hydrophobic surfaces [95]. In the touch transfer assay, a sterile velveteen cloth is wrapped on top of a cylindrical weight and the tied cloth is immersed in a known amount of pathogen. The excess liquid is drained out using another piece of cloth and the tied cloth is pressed on to the surface of the sample. In order to get the colony count, after transferring the pathogen on to the sample, it is pressed to a fresh agar plate on the sample surface and incubated at 37 °C for 24 h. As an alternative method of inoculation, a cotton swab charged with a known amount of bacteria can be used. | This test can be used to identify both antifouling and anti-microbial activity of coating under dry conditions. The test better mimics real-life conditions [95]. |

### 4.1. Antifouling Coatings

### 4.1.1. Nanostructured Surfaces

Adsorption of proteins onto a surface is the primary step leading to microbial adhesion; surfaces that can prevent this step are generally classified as self-cleaning or antifouling [86]. Natural self-cleaning phenomena, such as the water-pinning and water-rolling effects of rose petals and lotus leaves, are a result of the innate hydrophobicity of the surface [108]. The principle is applied to create self-cleaning coatings where super-hydrophobicity or hydrophilicity sweeps water from the surface along with any microbes, proteins, and other conditioning molecules. The unique design of these natural surfaces with topographies in the nanoscale is the reason for their superhydrophobic or superhydrophilic properties and the resultant self-cleaning properties. To mimic natural antifouling, material surfaces can be engineered with well-defined nano-topographies, resulting in anti-adhesive hygienic surfaces [109,110]. For these types of coatings, the self-cleaning property depends on the physical morphology or the topology of the surface. Examples include imprinting nano- or micro-patterns on a three-layer polypropylene polymer matrix using nanoimprint lithography [95] and fabrication of polydimethylsiloxane elastomers to mimic the skin of fast-moving sharks [107,111].

These types of coatings are ideal for medical devices that come into blood contact that need to resist protein biofouling [111]. The nanostructured surfaces contain no toxic chemicals [111]; however, the plastics used to make these surfaces are not eco-friendly, and the cost of disposal is high [112], but there is less wastage of materials, with a high output of uniform, compact, and stable products [112]. However, commercial installation of these surfaces in healthcare facilities is limited by cost, the current impracticality of creating nanostructure topographies for large surfaces, and the fact that damage to the surface is not easily repaired.

### 4.1.2. Chemically Modified Microbe Repelling Coatings

AFC can be prepared by tethering synthetic molecules onto surfaces to mimic natural self-cleaning phenomena [108]. Alumina, modified by an azobenzene type ligand and aromatic bis-aldehyde, creates a hydrophobic rose-petal-like surface with a water contact angle at 145°, and the surface exhibited a self-cleaning water pinning effect [108]. Teflon, siloxane, and fluorosiloxane surfaces are also superhydrophobic, which repel proteins and microorganisms that would otherwise attach to the surface [86].

Hydrogen bonding is important to protein fouling, and so antifouling surface surfaces should present no or low levels of hydrogen-bond donors/acceptors [113,114]. A typical protein-repellant coating contains polyethylene glycol (PEG), which resists protein adsorption [115]. Further, it has been shown that the detachment of pathogens on a normal substrate is difficult compared to the PEG surface showing the weak interaction between bacteria and PEG [115]. The properties of ethylene glycol, namely conformational flexibility, hydrophilicity, and the hydrogen bond-forming ability with water, are responsible for its protein-repelling activity [115].

Compared to nanostructure surfaces, PEG or PEG-based surfaces are easily repairable [112]. That means if the repelling coating is damaged, it is always possible to re-coat the surface to regain activity. However, the durability of these coatings toward weathering, chemical exposure, and mechanical stress is not well studied. From the literature search we conducted, it was observed that all self-repelling coatings are not suitable for large substrates or surfaces [112]. It is difficult to study antifouling activity under dry conditions; however, a recent article demonstrated the effectiveness of a sharklet micropattern in reducing the immediate contamination of a surface by *S. aureus* using a touch transfer protocol and was able to follow the reductions in bacterial burden at the surface through time. The sharklet micropatterned surface was superior to an antimicrobial copper surface for at least 90 mins of the experiment [107].

*4.2. Antimicrobial Coatings*

4.2.1. Contact-Active AMC

Contact-active AMC offers a safe by-design approach where the pathogen is killed without the release of antimicrobials from the surface. The coating can be very thin, down to a molecule-thick layer, meaning that little antimicrobial will be used, and so the coating may be considered eco-friendly. However, the active agent does need to be immobilized on the surface and present active antimicrobial groups. The biocidal mechanism requires contact and no leaching of the active agent, so continual killing activity is achieved unless the surface is blocked (e.g., by residual killed cells and other debris). Contact active coatings are primarily based on antimicrobial polymers as biocides. Hence, we can classify these coatings as natural (derived directly from nature or synthesized by mimicking natural compounds) or synthetic (synthesized from chemical monomers as a chemical product) antimicrobial polymers anchored on surfaces to form contact-active coatings.

(a)     Contact-active AMC containing biomimetic polymers

Coatings containing chitosan, a biocompatible polysaccharide composed of N-acetylglucosamine and D-glucosamine, may be considered as good contact-active AMC. Chitosan is a natural cationic polymer due to its positively charged amine groups [116]. Chitosan is an eco-friendly option for a biocide as its activity can be tuned by manipulating the amine functionalities and incorporating them in suitable polymers [117]. During biocidal action, the surface of a polycationic biocide will be covered with dead cells leading to loss of activity. However, washing these surfaces with cationic detergents can restore their activity [115].

An alternative category of antimicrobial substance that can be anchored at surfaces on polymeric brushes is the antimicrobial peptides (AMP) [118]. Antimicrobial peptides (AMP), such as magainin and defensin, or mimics, can be incorporated into contact-killing AMC [119]. The mode of action of AMP is fundamentally based on two properties, first, a highly rigid backbone, and second, the arrangement of one hydrophobic and one cationic side group that enable the AMP to penetrate and destroy the cell membrane [119,120]. In order for the surface-anchored AMP to be effective, high doses are required, which can be toxic as well as costly [86].

(b)     Contact active AMC containing synthetic polymers

A typical contact-active AMC, quaternary ammonium compound (QAC), forms electrostatic bonds with negatively-charged cell membrane components and disrupts the integrity and function of the whole cell [86]. At lethal concentrations, it displaces cytoplasmic components and dissociates the phospholipids [121]. Direct incorporation of QAC into a coating can lead to high levels of leaching, fast depletion of biocide, and high eco-toxicity. A more accepted method is to graft the active QAC with a polymer that can bind/anchor the QAC to different surfaces. [122]. The biocide 3-(trimethoxysilyl)-propyldimethyloctadecyl ammonium chloride, an organosilicon quaternary ammonium compound (Si-QAC) [122,123], is anchored by the Si-QAC silane group that condenses with free hydroxyl groups on the surface and stabilizes by intermolecular siloxane (Si-O-Si) linkages [124,125]. The antimicrobial mechanism is reliant on the quaternary amine (N+ atom) that attracts the negatively charged microbes onto the needle-like C18 structure of the hydrophobic chain, which leads to the puncture of the bacterial cell envelope and events that cause cell death [117].

Polyaniline (PANI) has been reported as a good contact-active surface material that can be used for AMC [126,127]. The PANI surface is conducting, and it attracts the negatively charged bacteria through an electrostatic force [128]. However, the change in the structural chemistry of PANI with pH (emeraldine salt form is more active than emeraldine base form) may pose a challenge to the sustainable antimicrobial activity [129,130].

The commercial application of contact-active AMC seems to be more difficult than the commercial application of their releasing counterparts since contact-active AMC requires more time and cost for fabrication. Furthermore, although they may be less environmentally damaging than releasing AMCs, the active agents may still be depleted.

4.2.2. Biocide-Releasing AMC

A surface that releases a biocide is a conventional route to the design of an AMC. A variety of broad-spectrum agents can be released, but a number of challenges do exist; for example, dead cells may remain and interfere with subsequent biocide activity, the reservoir of biocide may become exhausted, limiting the active life of the surface and promoting the development of resistance at sub-MIC levels, and the biocide released may have human and eco-toxicity [84,86,131]. Biocide-releasing AMC can be classified based on their action of biocide release.

(a) Continuous release

The simplest approach involves the continual release of biocide from the surface. The gradient of biocide formed will establish an outer inhibition zone and an inner kill zone to protect the surface against colonization by pathogens [84]. As cells are often killed before reaching the surface, they do not attach and may be easily cleaned [132]; however, these surfaces deplete at a rapid rate and may need to be replaced frequently. Generally, metals, metal oxides, and nanoparticles of Ag and Cu are used as biocides in these AMC.

The release of Ag from AMC has been accepted widely as a promising method for biocidal activity. The positively charged Ag ions accumulate at the predominantly negatively charged parts of the microbial envelope, causing cell membrane damage and consequent microbe killing [133]. The AMC with Ag as an active agent can be formulated by incorporating Ag alloy, Ag nanoparticles, Ag oxide, chelated Ag, metallic Ag, or Ag salts [117]. Among these, formulations containing Ag nanoparticles showed comparatively better activity since they can release Ag ions, generate reactive oxygen species (ROS), and damage the cell membrane directly [117,134]. Copper is a metal with similar biocidal properties to Ag that is relatively cost-effective and less toxic to humans and the environment. The ability of Cu to degrade/damage DNA effectively limits the transmission of plasmids containing antibiotic and biocide-resistance genes [4,135], although resistance to both Ag and Cu has been documented [136,137].

Cu and Ag coatings exhibit high levels of antibacterial and antiviral activity [128–131] and are easy to apply to different types of surfaces [138]. Cu and Ag can be coated on many surfaces, such as stands/poles carrying intravenous fluids, bed rails, call buttons, doorknobs, and other small objects, but have not been extensively tested for application on large areas, such as walls. Furthermore, the real-world effectiveness of these biocidal surfaces in reducing the risk of HAIs is still under investigation, and their durability inside a real-world hospital remains limited to date [70], although studies show promise for Cu coatings [139–143]. In those studies, CFU burden and viable cell reductions of the inocula are below internationally-accepted values (<250 aerobic CFU per 100 cm$^2$ of surface area; >99.9% reduction after 60 min) [144]. A major disadvantage of Ag coatings is the decline in activity observed at low temperatures and at low humidity levels that are often similar to real-world conditions [96,138]. It has been shown that even though Ag coatings are active when tested using JIS Z 2801-based protocols, they show no significant response under dry conditions, as they need moisture to work. Nevertheless, this is not the case with Cu, as these coatings show activity even under dry pathogen exposure conditions [96,138]. Another disadvantage is that scratches of the coating that result from wear may serve as safe hiding spots for pathogens, making Cu and Ag coatings to be less active than expected [138].

(b)  Slow release

Achieving a slow rate of biocide release can overcome the disadvantage of rapid exhaustion of biocide reservoirs in continuous release AMC. The technique is to incorporate the main biocide(s) in a suitable polymer matrix to control the rate of release.

Drug release techniques can be implemented in AMC-containing antibiotics. In order to affect slow release, it is important to entrap the biocide material in a suitable biocide carrier, such as polylactic acid or other polyesters [145,146] or polyelectrolyte multilayers [147]. Antibiotics can be used as suitable biocide candidates in slow-release AMCs, and in such coatings, hydroxyapatite (HA) is preferred as a suitable carrier due to its biocompatibility [148]. The drug release rate can be controlled by tuning the concentration and composition in the coating [148]. A modified form of HA, carbonated hydroxyapatite (CHA), showed better incorporation and slower release compared to HA coatings for antibiotics containing carboxylic acid groups, such as amoxicillin, cephalothin, carbenicillin, and cefamandole [149].

Those surfaces that use Ag without releasing the metal continuously may be classified under the slow-release AMCs. For example, glass slides coated with nanosized Ag in the 1–2 nm range and modified with highly branched amphiphilic poly(ethyleneimines) (PEI) showed significant activity against *E. coli* without releasing much Ag into the environment [150]. Control experiments carried out with PEI alone, PEI with silver nitrate mixture, and PEI with reducing agent lithium triethylborohydride showed no antimicrobial activity suggesting the important involvement of Ag nanoparticles [150].

Slow-release coatings will be expected to retain potency against target pathogens for a longer period of time when compared to continuous-release coatings. However, the relative cost of the coating will be higher than continuous releasing counterparts, and this is a major disadvantage when considering these coatings for application on large objects.

(c)  Triggered release

The continuous leaching of biocide from the AMC questions its efficiency and durability. Ideally, we want surfaces that can detect the presence of microbes and only release biocide when it is needed. Triggered-release biocidal coatings solve this problem by only releasing the biocide in response to specific external stimuli [86]. Examples of these stimuli include bacterial molecules and elements of the host's response to infection. Quorum-sensing molecules are released by bacteria as means of communication [151,152]. On a surface, the concentration of quorum-sensing molecules (e.g., homoserine lactones for Gram-negative bacteria) increases with the multiplication of the bacteria and beyond the threshold value, can be used to trigger the release of biocide [151]. An example includes the release of the antibiotic ciprofloxacin by the quorum-sensing lipase-sensitive homoserine group incorporated on the surface of a PEG-like polymer [153,154]. A second example responds to an increase in the protease thrombin, seen during a bacterial infection, to trigger biocide release via a thrombin-degradable peptide linker crosslinked with polyvinylalcohol, which then releases the encapsulated antibiotic [155].

These AMCs are more complicated than the other two types of releasing AMCs. Hence, the triggered-release AMCs may be used for special surfaces. The requirement of specific stimuli to activate the coating raises questions regarding the specificity of activation. Alternative approaches have been suggested that use a specific trigger that sense when to clean a surface by signaling the presence of infection/colonization or comprise a surface that can be externally triggered, for example, by temperature [156,157] or pH [158,159] when other evidence of colonization is apparent.

## 5. Antimicrobial Actions of Coatings with Both AMC and AFC

The problem with having only one type of coating (AFC or AMC) is that it can only perform one function, while both antimicrobial actions are needed to inactivate pathogens and antifouling performance to repel the dead/live pathogen from the surface to renew its activity. The combination of different AMC and AFC components is seen as an approach to overcome the limitations observed for coatings with a single component. The options are categorized in Figure 4. There are other types of AMC that are modified to be more active (e.g., a combination of releasing and contact-active AMC). Some coatings apply the ability of AFC to self-clean the surface to expose active sites again after killing [158]. The renewal of the surface after bacterial adhesion or killing provides enhanced synergistic activity for the AMC. However, even though it is interesting to study such combinations of different functional materials in coatings and these coatings look attractive in terms of activity, the cost and complicated preparation make them unsuitable for common or high-touch surfaces.

### 5.1. Contact-Killing and Repelling Coatings

As illustrated in Figure 4, these coatings kill pathogens upon contact and possess the ability to repel dead microbes. There are not many examples of this type of coating in the literature; however, a suitable example is a coating with Hydramacin-1 (HM-1) and lysozyme incorporated with a PEG-based spacer [160]. Here, HM-1 and lysozyme can act as contact-active agents, and PEG is a known microbe repellant.

### 5.2. Releasing and Repelling Coatings

A common approach toward this mechanism is found in multi-layer coatings. For example, a coating containing Ag nanoparticles embedded in polyelectrolytes, which are covered by a top layer of polyzwitterion, a 2-methacryloyloxyethyl phosphorylcholine (MPC) copolymer with 2-aminoethyl methacrylate (AEMA), which is antifouling in nature due to its high hydrophilicity [161]. In another example, a combination of a killing agent with an antifouling polymer, such as poly(N-isopropylacrylamide) (PNIPAAm), which is thermo-responsive in nature, can be employed for repel and kill dual function [159]. A wide variety of killing units have been reported to work synergistically with PNIPAAm, including QAC, AMP, lysozyme, antimicrobial polymers, and Ag nanoparticles [159]. The biocides were pre-immobilized between PNIPAAm brushes, and the temperature-responsive conformational changes exposed the active agents to the adhered bacteria [156,157]. Polymers that are sensitive to pH changes can also be used for such coatings. A typical example of such polymer is poly(methacrylic acid) (PMAA), which has several carboxylic acid-repeating groups [159]. This group releases protons under the basic condition of yielding carboxylate units with a negative charge that can repel bacterial cells electrostatically. The surface consists of a hierarchical two-layered structure where the hydrophilic PMAA with negative surface charge shields the biocide in the inner layer [162]. As the bacteria start to colonize the film, their metabolism reduces the local pH at the surface, resulting in the collapse of the PMMA layer and consequent release of the AMP [159]. The dead bacteria are removed by hydration, which increases the pH and results in the swelling of the polymeric layer [158].

### 5.3. Releasing and Contact-Killing Coatings

Introducing contact-killing agents to releasing type coatings or vice versa can significantly increase the durability and performance of the surface. Generally, these types of coatings have no self-cleaning ability, and dead microbes may remain on the surface to act as a shield for microbes above that may go on to form a biofilm on the surface or spread and cause HAIs. Not many examples of these coatings are reported in the literature, but one suitable example is a composite coating with Ag nanoparticles, p-Aramid antimicrobial fibers, and glycidyltrimethylammonium chloride has been reported to show superior activity than coatings with the individual components [163].

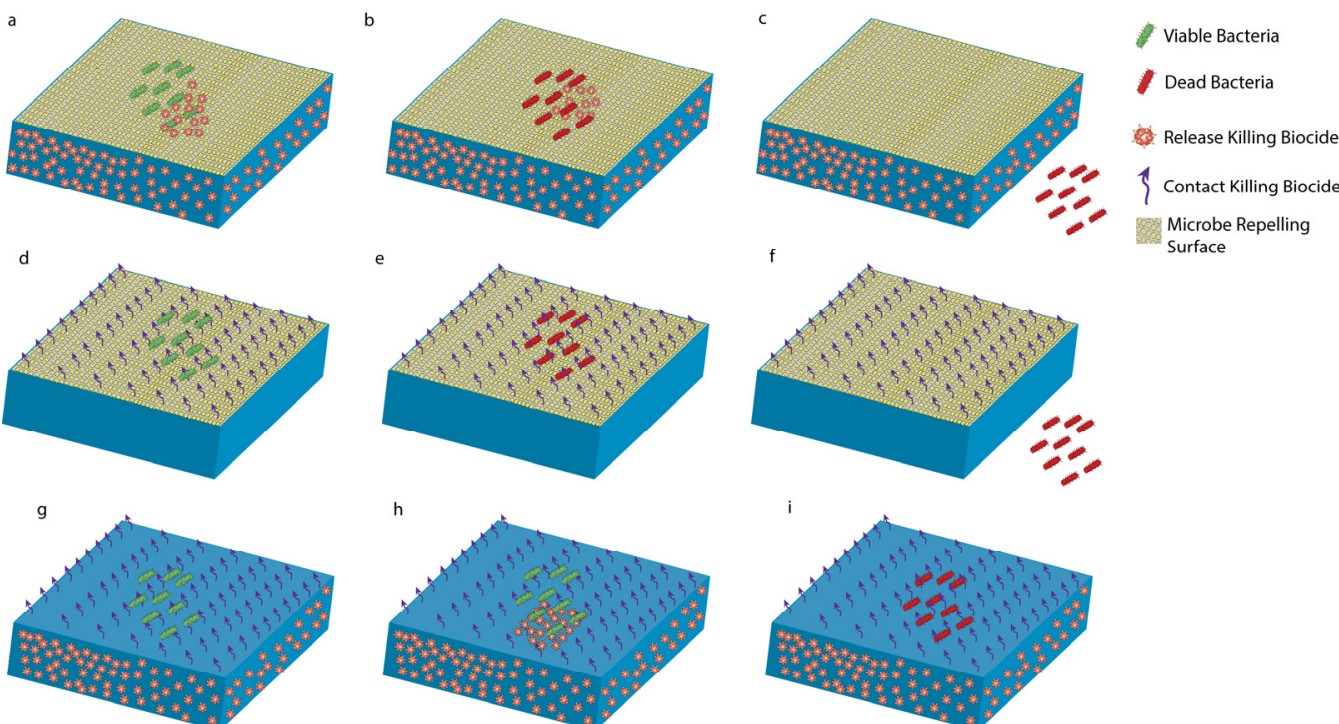

**Figure 4.** Types of antimicrobial coatings. Releasing and repelling coatings: (**a**) Bacteria encounter the surface. (**b**) Bacteria are killed by biocides released from the surface. (**c**) Self-cleaning surface removes the dead cells. Contact-killing and repelling coatings: (**d**) Bacteria encounter the surface. (**e**) Bacteria killed upon contact. (**f**) Self-cleaning surface removes the dead cells. Releasing and contact-killing coatings: (**g**) Bacteria encounter the surface. (**h**) Bacteria killed by both biocides. (**i**) Dead bacteria stay on the surface until cleaned.

## 6. Replenishable Coatings: A Sustainable Option?

AFCs, releasing AMCs, and contact-killing AMCs present application difficulties for sustained protection of surfaces, particularly for frequently touched hospital surfaces that may have a large area and can contribute to the spread of nosocomial pathogens. In the field of AMC, another category exists that possesses the property of rechargeability and offers the potential of being replenishable, and so overcoming the limitation of the exhaustion of the biocide reservoir. The examples of photocatalytic and N-halamine coatings can be considered as a special category of releasing coatings, which can be easily replenished.

### 6.1. Photocatalytic Coatings

The photo-disinfection properties of TiO$_2$, first identified by Matsunaga et al. in 1985, later became a platform for a new field of AMC research [164], forming biocide with water, oxygen, and light. The photocatalytic materials work through reactive oxygen species (ROS) produced at the pathogen photocatalyst interface upon irradiation with light of suitable energy (see Figure 5) [165]. Generally, a photocatalyst is a metal oxide semiconductor (e.g., TiO$_2$) with electrons in its valence band at the ground state. Upon irradiation, the valence band electrons are excited to the conduction band and create electron–hole pairs. The electron–hole pairs have high redox potentials to produce ROS from oxygen and water molecules (see Figure 5) [166]. The ROS produced photocatalytically include biocidal hydrogen peroxide, superoxide radicals, and hydroxyl radicals, which kill via several paths and attack different biological sites. This makes photocatalysis effective toward a broad spectrum of microorganisms, including antibiotic-resistant microbes [165,166]. However, the ROS concentration must be higher than the pathogens' tolerance/stress threshold in order to achieve a complete inactivation [165,166]. In most situations, ROS acts on bacteria from the outside, although there are exceptions, e.g., where photoactive substances, such

as Ag, are ingested by bacteria [167], and so the size and thickness of the outer cell wall of the pathogen determines the resistance to photocatalysis within the given time of exposure. Hence, comparing different classes of microorganisms, it generally takes a longer time to kill spores > molds > yeast > Gram-positive bacteria > Gram-negative bacteria > prions > viruses through photocatalysis alone [165,168].

An added feature is the photoinduced hydrophilicity of these coatings that imparts a simultaneous AFC behavior. For example, $TiO_2$-based coatings have an innate antifouling property because of this change in hydrophobicity, with AMC containing $TiO_2$ possessing light-activated superhydrophilicity and showing self-cleaning of dead pathogens. It was reported that UV irradiation of $TiO_2$ films enabled the coatings to be highly hydrophilic as well as oleophilic [169]. A contact angle of almost zero was achieved with both water and oily liquids after the UV irradiation [169]. UV light exposure to $TiO_2$ surfaces leads to oxygen vacancies and reduction in $Ti^{4+}$ to $Ti^{3+}$ to form sites favoring the dissociative adsorption of water. Therefore, after UV exposure, domains that are hydrophilic and oleophilic appear on the coating surface, and this high amphiphilicity is maintained after the removal of the UV light source [169]. Additional nano topographical surface fabrication of $TiO_2$ coating will achieve a multifunctional with enhanced self-cleaning and antimicrobial ability [104].

The major limitations of photocatalytic coatings are that no antimicrobial activity is seen in the dark, and activation requires a UV light source [170]. Materials added to the coatings can boost the photocatalytic activity at the lower-energy visible light wavelengths and provide antimicrobial activity in the dark [171]. Visible light activation of $TiO_2$ can be achieved by incorporating other photoactive materials or doping it with metals, such as Ag and Pt, or non-metals, such as S and N [165]. A much higher visible-light activity was observed when $TiO_2$ was doped with Cu (metal) and F (non-metal) [171]. In this case, Cu acts as a booster to enhance killing and promotes dark activity, and F acts as a dopant that allows activation of $TiO_2$ by visible light [171]. Another common alternative to Cu is Ag, and an Ag-modified photocatalyst can induce intracellular ROS production in addition to ROS produced extracellularly [165,167].

Photocatalytic coatings offer low-cost, non-toxic surfaces with high chemical, thermal and light stability, wide, large-scale availability, and light-induced self-cleaning properties with the ability to be incorporated into transparent coatings [104,172] and the potential to be used for any substrate that can be exposed to light. However, the stability against light is questionable in cases where it is combined with other compounds susceptible to ROS attack (such as a coating with $TiO_2$ and binder). In such a case, the major disadvantage is the self-oxidation of the coating upon UV weathering, which disintegrates the structure and compromises its activity (also known as photobleaching). However, in order to be used in a real hospital environment, a compromise on the lifetime of the photocatalytic coating upon continuous irradiation must be made. Photocatalytic titania can be implemented on environmental surfaces, medical implants, and medical devices [165,173]. The wide-range application of these coatings makes them attractive for the development of coatings for most of the surfaces in hospitals. Compared to copper and silver, which may corrode or be consumed over time, photocatalytic $TiO_2$ coatings may be preferred [174]. During the peak period of the COVID-19 pandemic (2020–2021), a Finnish company Nanoksi successfully marketed a long-lasting and efficient nano-$TiO_2$-based coating product for airports and other commercial installations in Europe and UAE [175,176].

The possibilities of developing new $TiO_2$-based photocatalytic coatings with high activity under visible light, dark conditions, and enhanced self-cleaning action are numerous and have stimulated research to look further into the photocatalytic AMC. Limitations still exist, in particular, around how the low level of photocatalytic activity under dry conditions or low humidity conditions, which relates to the real-world environment, can be improved or augmented with additional antimicrobials.

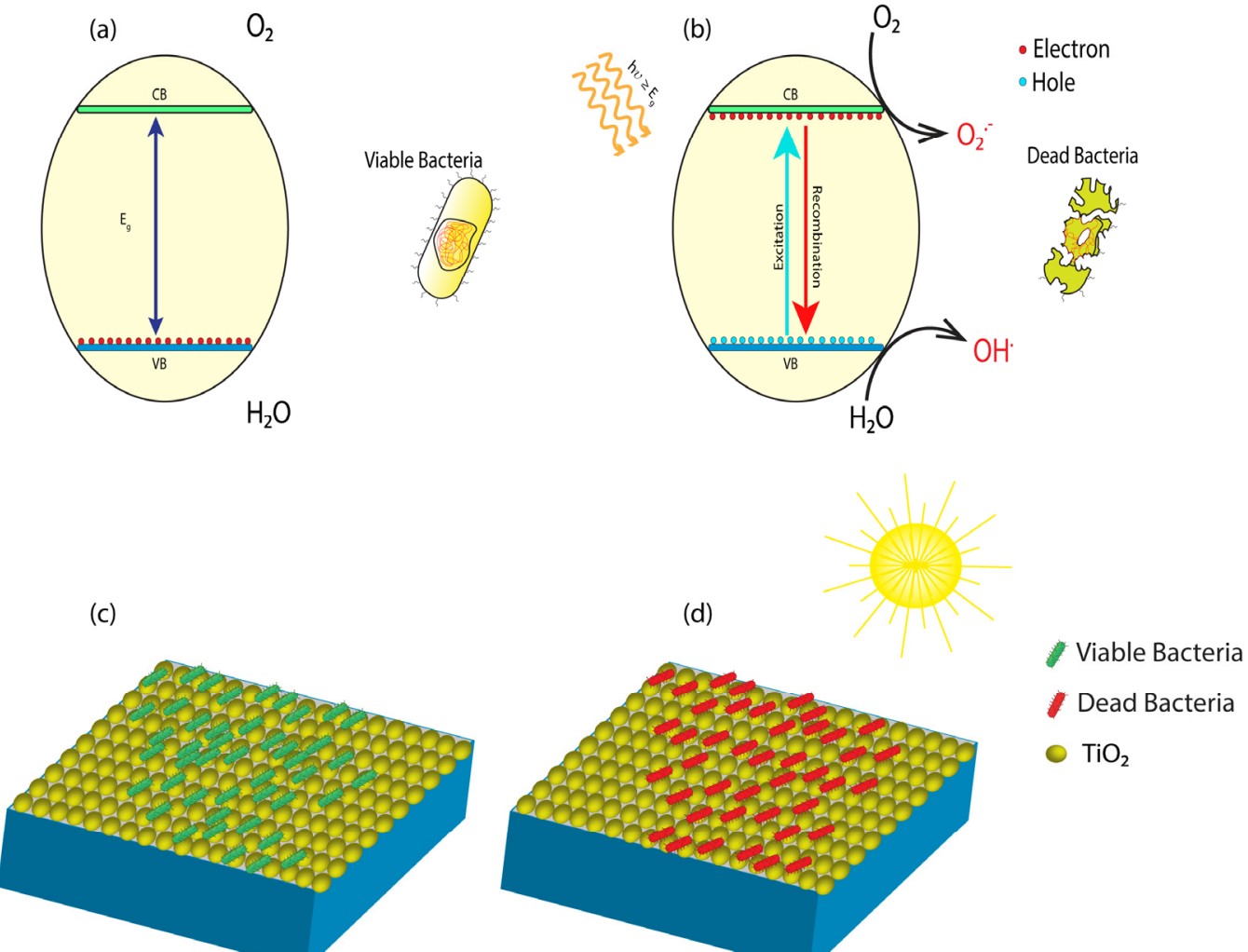

**Figure 5.** Photocatalytic surfaces. Using the example of a semiconductor photocatalyst with a valence band (VB) and conduction band (CB). (**a**) In the absence of light, ROS are not produced, and viable bacteria are unaffected. (**b**) Upon irradiation with light having sufficient energy, electrons in VB are excited to CB creating active electron–hole pairs that produce biocidal ROS (hydroxyl OH$^.$ and superoxide O$_2 \cdot^-$ radicals), thereby killing the bacteria, (**c**) Live bacteria on TiO$_2$ coated surface in the absence of light, and (**d**) dead bacteria after illumination.

### 6.2. N-halamine Coatings

Even though photocatalytic coatings are highly suitable for most hospital surfaces, the time to kill pathogens depends on their ROS susceptibility and the intensity of activating light [165,168]. There are surfaces that need quick disinfection, and a coating with broad activity is demanded for these surfaces. For example, face masks need a significant and rapid antimicrobial action [177]. In the case of surfaces, such as a face mask, the frequency of touches is high, and this proportionally increases the probability of acquiring and transmitting pathogens by hand if they are not killed quickly [177]. Additionally, the negative effect of the photocatalytic component on the light stability of the whole coating demands a longer-lived AMC. Coatings that contain N-halamines may be a solution to achieve superior long-life antimicrobial activity against a broad spectrum of pathogens [177].

An N-halamine is a compound that contains a chemical structure where the nitrogen in the molecule is chemically bonded with a halogen atom (Cl, Br, or I, with Cl being the most popular) [178]. The mechanism of formation of a biocidal N-halamine system from a non-biocidal precursor (any >N-H group containing molecule) is illustrated in Figure 6. In simple terms, the >N-H group is halogenated or converted to >N-Cl group by simple treatment with a dilute-bleach solution. The halogen is released into the aqueous system in its +1 oxidation state, which possesses powerful antimicrobial potency against a broad spectrum of pathogens [178–180]. In these compounds, the active biocides are the free halogen species that are mostly released slowly but can undergo rapid release in some cases. The active halogens are covalently bonded to nitrogen to impart stability and ensure controlled release. The N-halamine works by direct transfer of oxidative halogen to the biological molecule (see Figure 6) [181,182]. The oxidative halogens in the form of Cl+ or Br+ attack thiol groups and amino groups in proteins, which leads to functional inactivation and cell death when sufficient proteins are inactivated [183,184]. After biocidal action, the N-halamine molecule reverts to its original state (>N-Cl reverts to >N-H). The long-lasting biocidal property of this AMC arises from the facile method of recharging the non-biocidal system to biocidal N-halamine, e.g., using a dilute solution commercial bleach for >N-Cl [178,185,186] (see Figure 6).

N-halamine coatings provide for AMCs with broad spectrum activity, non-toxicity, low cost, eco-friendly nature, ease of application, and rechargeability [177,178]. In particular, N-halamine coatings offer the advantage of a surface that can be cleaned and recharged by using a dilute-bleach solution (see Figure 6). Compared to photocatalytic coatings, N-halamine components do not promote the photobleaching of the coating components. Even though these advantages make N-halamines highly appealing, there are disadvantages regarding the stability of >N-Cl bond and other factors that have negative effects on the inherent antimicrobial properties.

The stability of any N-halamine AMC primarily depends on the N-halamine component present. Hence, it is important to discuss the factors affecting the stability of various N-halamine moieties. Before discussing the factors that affect the stability of N-halamines, a clear picture of its different molecular forms must be established. Based on the whole chemical structure, N-halamines can be generally classified as cyclic (N within a ring structure) [182] and acyclic (N present in the linear structure) [187] molecules. While considering N-halamine functional groups within a molecule, N-halamines can be present as primary or secondary amines, amides, or imides [178]. The molecular size of the N-halamine moiety and the adjacent groups present with >N-Cl functional group can further contribute to the stability of the N-halamines.

The key challenge for any N-halamine molecule is to limit the decomposition of its structure under light with a UV component. When compared to photocatalytic coatings, the N-halamine components do not degrade or boost degradation of the coating but instead, the biocide itself degenerates. N-halamines used in AMC in hospitals are likely to be continuously exposed to two types of light sources: indoor lighting inside the hospitals and sunlight. In both sources, there will be a UV component in its spectrum (UV intensity higher for sunlight). The unchlorinated form of N-halamine (>N-H group present) is stable toward UV light; however, the chlorinated/active form (>N-Cl group) will decompose when exposed to UV light, and all bound active chlorine is lost within hours [178,182,188–190]. Furthermore, the N-halamine molecule can undergo photolytic rearrangements upon UV exposure, thereby declining the rechargeability and the lifetime of the N-halamine coating [178,182,191].

Likewise, the biocidal efficiency, lifetime, and rechargeability are also affected by indoor light [178,192], heat [178,185,193], pH [178,194,195], hydrolysis [178,196–198], and chemicals (reducing and dechlorinating agents) [178,199–201]. A detailed consideration of the chemistry associated with each of these factors is beyond the scope of this review. Instead, we have ranked the stability of N-halamines toward these factors in Table 2. Similar to the release-type coatings discussed earlier (Section 4.2.2), N-halamines may suffer from

the accumulation of dead microbes, which may provide a safe zone for new microbes. Two more key questions regarding the real-world activity of N-halamine coatings remain: (1) can they be charged by simple wiping with a dilute-bleach solution to activate N-halamines applied on larger areas, and (2) will they show antimicrobial activity in both wet and dry conditions? From our literature search, we have only found papers reporting chlorination of N-halamines by immersion of the samples in a bleach solution and the activity for surfaces tested under wet conditions.

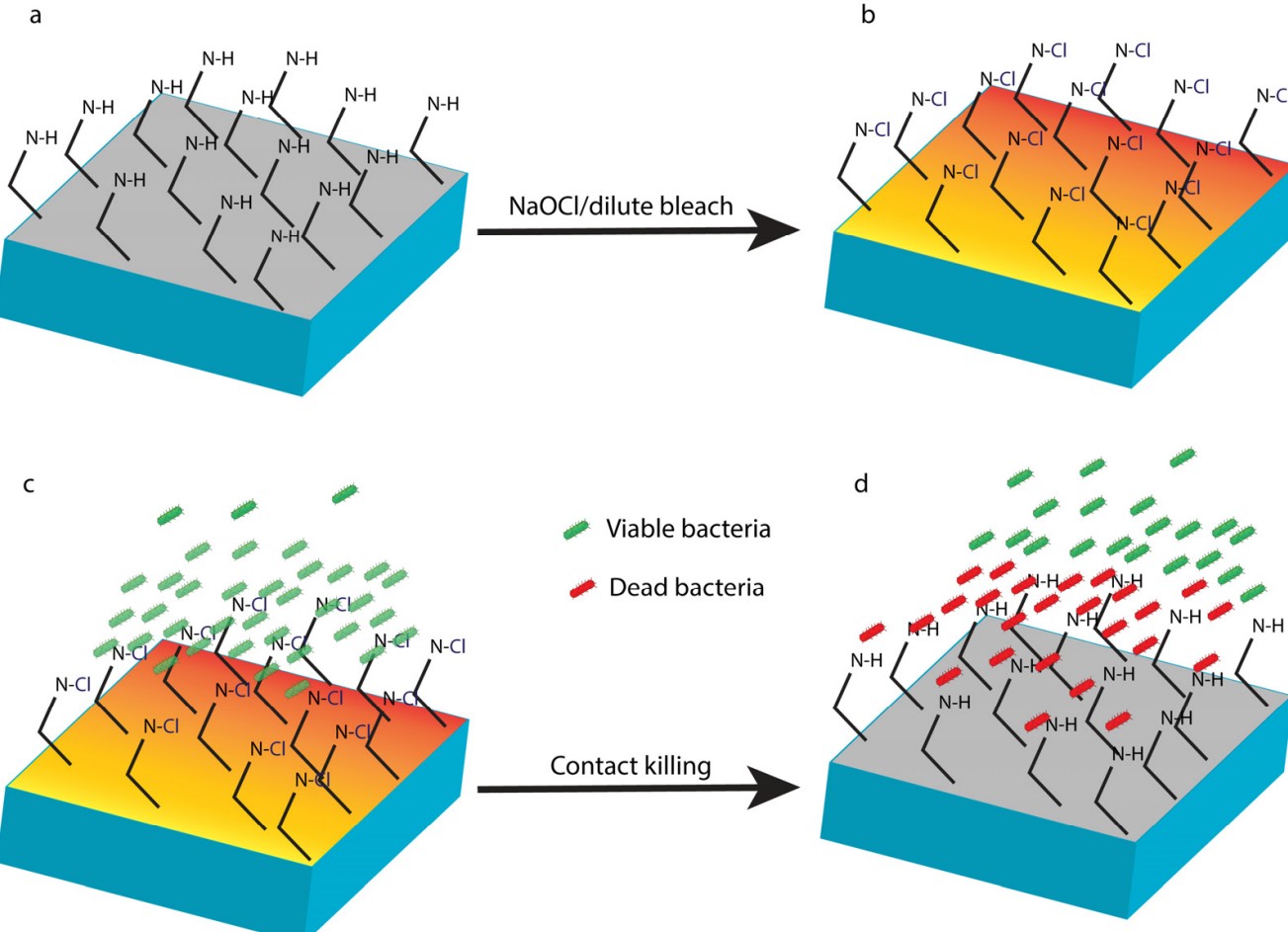

**Figure 6.** N-halamine coatings. A coating with a polymeric N-halamine precursor with >N-H functionalities on the surface (**a**) undergoes functionalisation of the surface amine groups with a dilute-bleach solution or NaOCl solution to form biocidal >N-Cl groups (**b**). Live bacteria encounter the surface (**c**) and are killed upon contact with >N-Cl groups (**d**), noting that after killing, the surface returns to (**a**) and becomes inactive until the next recharge cycle.

**Table 2.** Ranking the common N-halamine molecules based on stability and activity.

| N-halamine Type | | Structural Stability [178,202] | UV Light Stability [178,182] | Indoor Light Stability [178,192] | Water/Hydrolysis Stability [178,182,196,197] | Antimicrobial Activity [178,182,203–205] |
|---|---|---|---|---|---|---|
| Chemical structure | Cyclic | * | High | High | High | * |
| | Acyclic | * | Low | Low | Low | * |
| Functional group | Amine | High | * | * | High | Low |
| | Imide | Low | * | * | Low | High |
| | Amide | Moderate | * | * | Moderate | Moderate |
| | Multiple >N-Cl bonds | * | * | * | * | Highest |
| >N-X | F | * | * | * | * | * |
| | Cl | * | * | * | * | Low |
| | Br | * | * | * | * | Moderate |
| | I | * | * | * | * | High |

* Not reported.

## 7. Selecting Suitable Coatings for Hospitals: A Major Challenge?

A number of coatings have been reported that can kill and repel pathogens, but are these coatings suitable to tackle the HAI problem? To answer this, the coatings should be active against a broad spectrum of nosocomial pathogens, be non-toxic to humans and the environment, exhibit durable, sustained activity, and be active in real-world trials.

Testing a broad spectrum of microorganisms is a time-consuming task, and many studies will be satisfied by testing a Gram-positive and a Gram-negative bacterium. More exhaustive studies will test more diverse example species, including antibiotic-resistant isolates, bacterial endospores, and viruses. The emergence of COVID-19 has encouraged substantially more testing of antiviral activities. Fewer articles report the durability of antimicrobial surfaces with respect to simulated UV weathering, disinfectant cleaning, temperature, humidity, or time. Often, articles also fail to assess the adhesion of coatings to the range of surface types it might reasonably be used upon, which may influence the durability of the coating itself. Among the examples given in Table 1, only one publication, [206], studied the durability of the coating against mechanical stress. In addition, while the research will often investigate the cytotoxicity of a surface, much fewer reports exist where an assessment of eco-toxicity is made.

So, although many possible antimicrobial surface options can be proposed, there are much fewer studies that investigate the "fitness for purpose" where there is an evaluation of durability and sustained activity for activity against the key nosocomial pathogens, including drug-resistant bacteria, endospores, fungi, and viruses.

It is obvious from Table 3 that there are numerous coatings that show good antimicrobial activity, with the potential for application in a hospital to tackle HAIs. Unfortunately, there is no specific single standard test for assessing the acceptable efficacy of antimicrobial coatings used in hospitals. Perhaps the best available protocol is produced by the U.S. Environmental Protection Agency (EPA), which details the mandatory conditions for all antimicrobial coating manufacturers and may be considered as a satisfactory standard for hospital coatings at the moment [97,207–209]. EPA suggests that an effective product should show at least a three-log reduction within 1 h after inoculating with respect to control carriers [144]. If we apply this standard to Table 3, only the cupric oxide coating [210] will pass. Most papers fail to report the acceptability of a coating in terms of testing with standards for durability, such as adhesion tests on different substrates, UV weathering, stability toward temperature and humidity, and durability after cleaning with standard disinfectants.

**Table 3.** Efficacy of self-disinfecting coatings toward critical antimicrobial pathogens.

| Type of Surface | Active Component | Conditions | | Tested Pathogens | Activity (Microbe Repelled/Killed) |
| | | Inoculum | RH, T and t | | |
| --- | --- | --- | --- | --- | --- |
| Nanostructured AFC | Superhydrophobic surface integrated with micro-pillar arrays and packed nanoneedles [211] | 2 mL $10^8$ cfu/mL | 37 °C 24 h | *E. coli* | >99%* |
| Chemically modified AFC | PEG [212] | 5 µL $10^8$ cfu/mL | 37 °C 2 h | *S. aureus* *E. coli* | 90%* 90%* |
| Continuous releasing AMC | Cupric oxide [210] | 5 µL $10^7$ TCID$_{50}$/mL | 60−70%, 22−23 °C 30 min | SARS-CoV-2 | 99.8% |
| Slow releasing AMC | Silver and Thymol in poly(lactic acid) films [213] | 100 µL $10^4$ cfu/mL | 24 °C, 3 h | *S. aureus* *E. coli* | 47.5% 40.6% |
| Triggered releasing AMC | pH responsive poly(methacrylic acid) with antimicrobial peptide [214] | 250 µL $10^7$ cfu/mL | 1 h | *S. aureus* *E. coli* *P. aeruginosa MRSA* | 99.9% 99.9% 99.9% 99.9% |
| Contact active AMC | Quaternary ammonium polymer coating [215] | 50 µL | 30–50%, 22–23 °C, 2 h | SARS-CoV-2 Human coronavirus 229E | >3 log >5 log |
| Contact-killing and repelling coatings | Nano silica and fluorosilane with Lysozyme (muramidase) [206] | 1 mL/cm$^2$ of 6.3 and 6.6 log cfu/mL | 150 rpm, 24 h | *Listeria innocua* *Salmonella* Typhimurium LT2 | 4 log# 6.5 log# |
| Releasing and repelling coatings | Copolymer brushes of 2-hydroxyethyl methacrylate and 3-(acrylamido)phenylboronic acid with Quercetin [216] | 500 µL $10^7$ cfu/mL | 37 °C, 4 h | *S. aureus* *P. aeruginosa* | >80% >80% |
| Photocatalytic coatings | Melon/TiO$_2$ [217] | 20 µL $10^6$ cfu/mL | Actinic light 3 h | *S. aureus* | 99.9% |
| Rechargeable coatings | Polypropylene grafted methacrylamide [218] | 10 µL $10^7$ cfu/mL 10 µL $10^7$ pfu/mL | 15 min for bacteria and 5 min for virus | *L. innocua* *E. coli* T7 phage | >5 log >5 log 7 log |

Note: RH, T, and t: indicate relative humidity, temperature, and time. * indicates Repelling activity, # indicates reduction in bacterial numbers due to reduced adhesion and reduced growth, values not followed by * or # indicate killing.

In conclusion, there is a clear need for antimicrobial surfaces to limit fomite mediated spread of pathogens, especially in response to antibiotic resistance and viral pandemics. A variety of antimicrobial chemistries have been found to be effective in vitro, but many are either not suitable for larger-scale, real-world applications or have not been suitably tested. In particular, the key properties of durability and sustained activity need more exhaustive testing to determine whether a coating or surface treatment is fit for the purpose. In the future, we hope to see both a focus on the creation of surfaces with sustained, long-term activity and the development of testing standards that allow for a robust demonstration of sustained activity in real-world conditions, including after regular cycles of routine disinfection and recharging cycles to replenish the antimicrobial at the surface.

**Author Contributions:** All authors contributed to conceptualization, methodology, writing—original draft preparation, and writing—review and editing. Supervision, S.S. and M.G.-N.; funding acquisition, S.S. All authors have read and agreed to the published version of the manuscript.

**Funding:** This research was funded by the New Zealand Ministry of Business, Innovation, and Employment "The Biocide ToolBox for New Zealand Manufacturing Exporters" (UOAX0812) and a Callaghan Innovation PhD Scholarship RESE1902.

**Data Availability Statement:** Not applicable.

**Conflicts of Interest:** The authors declare no conflict of interest.

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
