# Peer review of "Antimicrobial Coatings: Reviewing Options for Healthcare Applications"

_2673-8007, doi:10.3390/applmicrobiol3010012_

Round 1

Reviewer 1 Report

OVERVIEW

The authors address the issue of the importance of surface contamination and transmission of microbes. In the first part, the authors present an overview of the problems regarding hospital-acquired infections. Subsequently, the authors point to the evidence for surface and fomite-mediated transmission in healthcare facilities and how fomite-mediated transmission can be managed. In the next part, the authors deal with surfaces (coatings) that help solve problems with fomite. Then the application possibility of antimicrobial actions of coatings with both antimicrobial coating and antifouling coating is indicated. The authors suggest an answer to the hypothesis: replenishable coatings are a sustainable option. Finally, the authors review the challenges and progress in developing an antimicrobial surface that can eliminate or at least repel pathogens. The authors used appropriate graphic elements that increase the quality and comprehensibility of the issue. The total number of references is above average.

CONCERNS

I consider the review to be of good quality. My comments are merely editorial (of minor type).

Minor concerns

1. The mark indicating the degree must be placed in the superscript position. The same applies to the degree Celsius mark. Correct accordingly throughout the text.

CONCLUSION

I find this review helpful. Regretfully, the paper cannot be accepted in its present form. The authors of the present review have to correct the issues.

Author Response

Thank you for the supportive evaluation. The degree sign code 00B0 has been used in place of superscripted o in the revised document.

Reviewer 2 Report

This paper reviews antimicrobial coatings for healthcare applications. It is very well-written, informative, well-structured, and easy to follow. The content is scientifically sound with exciting discussion. I would highly recommend this paper for publication. To add merit to the report, it is suggested to overview some available review papers on the field and state why this one brings something new to the literature. In addition, a "future outlook" section maybe required to show readers of the potential research direction.

Author Response

We thank the reviewer for their supportive assessment.

The reviewer has asked us to:

"overview some available review papers on the field and state why this one brings something new to the literature."

This is a good question that we have addressed in section 4, from line 162, where we have added new text, and some additional references. We have opted for a brief summary here highlighting what are broad reviews from 5-15 years ago, and some more recent reviews focusing on new/novel approaches. 

"The topic of antimicrobial surfaces and coatings for the purpose of reducing HAIs has been reviewed by others, covering antifouling, and biocide presentation and release options for self-disinfection [3,84-87] and focusing on the potential of new polymers and rechargeable antimicrobial chemistries [88,89]. In this review we provide updated classifications, with examples, of the variety of approaches to produce antimicrobial and antifouling surfaces. We highlight the limitations of some approaches and finally focus on the potential for surfaces that can provide sustainable and rechargeable activity for long-lasting protection."

The reviewer also asks a similar question to Reviewer 3:

In addition, a "future outlook" section maybe required to show readers of the potential research direction.

In re-reading the review we felt that the final section 7. Selecting suitable coatings for hospitals: a major challenge? attempts to include some future outlook. We have added text at the end of this section to emphasise this, choosing to be brief to leave the reader with a succinct concluding section.

"In the future we hope to see both a focus on the creation of surfaces with sustained, long-term activity, and the development of testing standards that allow for a robust demonstration of sustained activity in real world conditions, including after regular cycles of routine disinfection and recharging cycles to replenish the antimicrobial at the surface."

Reviewer 3 Report

If Figure 1 can be improved with better resolution, it will enhance the quality of paper.

The language of the manuscript needs to be more comprehensive and lucid for researchers new to the field.

Authors need to enhance the literature grammatically.

Authors need to summarise the literature and conclude the same for better readability of article.

Authors may wish to add future research perspective in the field.

Author Response

We thank the reviewer for their careful reading of the review.

The reviewer comments:

"If Figure 1 can be improved with better resolution, it will enhance the quality of paper."

We think the image in the template form is of lower resolution than the higher resolution image also submitted. We will ensure an image of the necessary quality is used in the final version. 

"The language of the manuscript needs to be more comprehensive and lucid for researchers new to the field."

"Authors need to enhance the literature grammatically."

It is always possible to improve the English language, and we have read through making some small changes to hopefully improve grammar. We have chosen not to add more details (to be more comprehensive) to hopefully focus on concepts illustrated by selected examples. We feel that the structure of the article leads the reader through the key points in a clear and logical order often offering headings that are key questions of value to experts in the field and newcomers. Reviewers 1 and 2 have commented/scored favourably on the readability of the article and we chosen not to attempt a wider ranging edit. 

"Authors need to summarise the literature and conclude the same for better readability of article."

The abstract summarises the content. We have tried throughout the article to summarise key questions and conclusions, successes or limitations at the start and end of each major section. We have chosen not to add further summary of the literature cited within the main text of each section as we prefer this more succinct approach.

"Authors may wish to add future research perspective in the field."